# Past COVID-19: The Impact on IVF Outcomes Based on Follicular Fluid Lipid Profile

**DOI:** 10.3390/ijms24010010

**Published:** 2022-12-20

**Authors:** Natalia Lomova, Natalia Dolgushina, Alisa Tokareva, Vitaly Chagovets, Natalia Starodubtseva, Ilya Kulikov, Gennady Sukhikh, Vladimir Frankevich

**Affiliations:** 1National Medical Research Center for Obstetrics Gynecology and Perinatology Named after Academician V.I., Kulakov of the Ministry of Healthcare of Russian Federation, 117997 Moscow, Russia; 2Department of Obstetrics, Gynecology, Perinatology and Reproductology, Institute of Professional Education, Federal State Autonomous Educational Institution of Higher Education I.M. Sechenov First Moscow State Medical University of the Ministry of Health of the Russian Federation (Sechenov University), 119991 Moscow, Russia; 3Moscow Institute of Physics and Technology, 141700 Moscow, Russia; 4Laboratory of Translational Medicine, Siberian State Medical University, 634050 Tomsk, Russia

**Keywords:** SARS-CoV-2, COVID-19, IVF, follicular fluid, metabolomics, lipidomics, mass spectrometry, biomarkers

## Abstract

Follicular fluid is an important component of follicle growth and development. Negative effects of COVID-19 on follicular function are still open. The aim of this work was to study the features of the lipid profile of follicular fluid and evaluate the results of the in vitro fertilization (IVF) program in women after COVID-19 to identify biomarkers with prognostic potential. The study involved samples of follicular fluid collected from 237 women. Changes in the lipid composition of the follicular fluid of patients who underwent COVID-19 in mild and severe forms before entering the IVF program and women who did not have COVID-19 were studied by mass spectrometry. Several lipids were identified that significantly changed their level. On the basis of these findings, models were developed for predicting the threat of miscarriage in patients who had a severe course of COVID-19 and models for predicting the success of the IVF procedure, depending on the severity of COVID-19. Of practical interest is the possibility of using the developed predictive models in working with patients who have undergone COVID-19 before entering the IVF program. The results of the study suggest that the onset of pregnancy and its outcome after severe COVID-19 may be associated with changes in lipid metabolism in the follicular fluid.

## 1. Introduction

In 2019, the world was shocked by the pandemic caused by the new coronavirus SARS-CoV-2 (severe acute respiratory syndrome-related coronavirus 2). To date, many questions remain about the impact of the SARS-CoV-2 on both reproduction and the course and outcomes of pregnancy [1,2,3,4,5]. Reproductive studies have revealed the absence of SARS-CoV-2 viral RNA in the follicular fluid (FF) or vaginal secretions of women with COVID-19 [6,7]. One of the studies conducted revealed specific antibodies to the virus not only in blood serum but also in ovarian follicles after an infection or vaccination [8], while the question of the adverse effect of COVID-19 on the function of follicles is still open. Metabolic studies of the FF of women infected with SARS-CoV-2 are of great potential scientific and practical interest. Many questions related to the metabolic pathways of COVID-19 disease remain to be resolved. SARS-CoV-2 infection is not fully understood at the level of cellular metabolites [9,10,11].

Follicular fluid is an important component of follicle growth and development and consists of many substances secreted by granulosa and theca cells, which transudate from the bloodstream. Changes in FF affect the developing oocyte [12], its maturation, the subsequent development of the early embryo, and the potential for its implantation [13,14].

The study of such signaling molecules as lipids in FF by mass spectrometry can give a new idea of the composition and functions of the follicular environment, as well as open up new links in the pathogenesis of a number of diseases leading to infertility [15].

It is known that patients with severe COVID-19 are exposed to a “cytokine storm” in the body, which causes a systemic inflammatory response and can damage any organs and systems, including the reproductive one [16,17]. Therefore, changes in the plasma composition can also be reflected in the FF composition. In addition, several biologically active substances released during inflammation modulate lipid metabolism [18], which indicates a possible relationship among the severity of COVID-19, the detection of certain metabolites, and the outcome of IVF.

The aim of this work was to study the features of the lipid profile of follicular fluid and evaluate the results of the IVF program after suffering COVID-19 to identify biomarkers with prognostic potential.

## 2. Results

### 2.1. Influence of Anamnesis

Taking into account the significant contribution of clinical data for further interpretation of the features of the FF lipid profile, the lipid levels were assessed depending on the anamnestic data, such as past COVID-19 and the severity of its course. The Kruskal–Wallis test revealed 10 lipids in the positive ion mode and 23 lipids in the negative ion mode (Appendix A, Figure 1), characterizing differences in FF associated with the history of COVID-19, belonging predominantly to the (lyso)phosphatidylcholine class.

At the same time, when comparing groups in pairs using the Mann–Whitney test, statistically significant differences between the group of healthy patients and patients who had a mild form of COVOD-19 were observed in four lipids in the positive ion mode and nine lipids in the negative ion mode (Appendix A). Changes were characterized by a decrease in the level of phosphatidylcholines and phosphatidylethanolamines in the case of anamneses of a mild form of COVID-19.

In a pairwise comparison of groups using the Mann–Whitney test, statistically significant differences between the group of healthy patients and patients who underwent a severe form of COVID-19 were observed in four lipids in the positive ion mode and eight lipids in the negative ion mode (Appendix A). Changes are characterized by an increase in the level of sphingomyelins, diacylglycerols, (lyso)phosphatidylcholines, and phosphatidylethanolamines in the case of severe COVID-19.

Statistically significant differences between groups of patients with different patterns of COVID-19 were observed in 26 lipids in the positive ion mode and 34 lipids in the negative ion mode (Appendix A). Changes were characterized by an increase in the level of cholesterol esters, (lyso)phosphatidylcholines, phosphatidylethanolamines, sphingomyelins, and cardiolipins in the follicular fluid in the case of a more severe course of the disease.

### 2.2. Predictive Models for the Threat of Miscarriage and the Success of an IVF Program Based on the Lipid Profile of the FF

From the combined set of lipids, lysophosphatidylcholines 16:0 and 18:1, plasmanyl-phosphatidylcholine 16:0/20:4, plasmenyl-phosphatidylethanolamine 16:0/20:4, and cholesterol ester 18:3 were selected as markers for diagnosing threatened miscarriage (Figure 2). The resulting model had an accuracy of 96% (CI 76–100%), a sensitivity of 95% (CI 65–93%), and a specificity of 93% (CI 59–100%) at a cutoff value of 0.48 (CI 0.01–0.99) (Table 1).

Of interest is the increase in the 16:0 concentration of lysophosphatidylcholine in the model. Lysophospholipids are precursors of arachidonic acid and act as second intracellular messengers. This cascade of transformations leads to tissue inflammation and impaired hemostasis.

Consideration of separate groups of patients with a mild form of COVID-19 and those without COVID-19 did not allow us to build a model for diagnosing a threatened miscarriage.

When considering a group of patients who had a severe form of COVID-19, cholesterol ester CE 20:5 and monogalactosyldiacylglycerol MGDG 18:0_18:0 were identified as markers of miscarriage (Figure 3). The resulting model had an accuracy of 99% (CI 98–100%), a sensitivity of 99% (CI 98–100%), and a specificity of 99% (CI 98–100%) at a cutoff value of 0.50 (CI 0.45–0.56) (Table 2).

It should be noted that in the resulting miscarriage prediction model, there was a significant increase in the relative concentration of cholesterol ester CE 20:5. Nicotra et al. showed that free fatty acid levels were significantly higher in women with recurrent abortion. The authors pointed out that an increase in the concentration of these molecules in women with recurrent miscarriage probably led to a stress-dependent release of catecholamines into the blood, followed by uterine vasoconstriction and placental ischemia. These mechanisms, combined with additional damage caused by reoxygenation, led to miscarriage [19].

Of particular interest was the creation of a model for predicting the success of IVF, depending on molecular, cytological, and clinical data.

From the combined set of lipids and cytological parameters, the number of blastocysts of the highest quality (BHQ), the number of mature oocytes, fertilization and blastulation parameters, phosphatidylcholines 12:0_22 and 18:0_20:3, plasmanyl-phosphatidylcholine 16:1/20:4, and plasmanyl-phosphatidylethanolamine 16:0/20:4 were selected as markers for predicting IVF success (Table 3, Figure 4). The resulting model is characterized by an accuracy of 70% (CI 42–97%), a sensitivity of 80% (CI 43–100%), and a specificity of 68% (CI 30–100%) at a threshold value of 0.30 (CI 0.01–0.64) (Table 3).

In a separate study of a group of patients who did not have COVID-19, phosphatidylcholine 16:0_16:1, plasmanyl-phosphatidylcholine 16:1/20:4, lysophosphatidylcholine 18:2, and plasmenyl-lysophosphatidylethnolamine 18:0 were classified as markers of the IVF success (Figure 5). The resulting model was characterized by an accuracy of 81% (CI 54–100%), a sensitivity of 94% (CI 68–100%), and a specificity of 77% (CI 45–100%) at a cutoff value of 0.28 (CI 0.01–0.63) (Table 4).

The model for predicting the success of the IVF program among patients who did not have COVID-19 included lipids of the phosphatidylcholine class, such as phosphatidylcholines 16:0_16:1 and plasmanyl-O-16:1/20:4, lysophosphatidylcholine 18:2, and lysophosphatidylcholine P -18:0. The resulting model had an accuracy of 81% (CI 54–100%), a sensitivity of 94% (CI 68–100%), and a specificity of 77% (CI 45–100%) at a cutoff value of 0.28 (CI 0.01–0.63). In cases of IVF failure, an increase in the relative concentration of lysophosphatidylcholine 18:2 was registered. It can probably be assumed that one of the leading triggers adversely affecting the process of embryo implantation, the early stages of its development, and, as a result, termination of pregnancy is the activation of a systemic inflammatory response through the production of arachidonic acid.

In a separate study of a group of patients who underwent mild COVID-19, the degree of blastulation, plasmanil-phosphaidylcholine 16:1/18:1, plasmenil-phosphaidylcholine 16:0/18:2, and phosphatidic acid 22:6_22:6 were classified as markers (Figure 6). The resulting model was characterized by an accuracy of 88% (CI 60–100%), a sensitivity of 96% (CI 68–100%), and a specificity of 85% (CI 51–100%) at a threshold value of 0.32 (CI 0.01–0.80) (Table 5).

In patients with failed IVF programs who underwent mild COVID-19, an increase in the level of plasmenil-phosphatidylcholine P-16:0/18:2 was registered. It can be assumed that IVF failures in mild cases of COVID-19 are not directly related to past infection and are probably due to other reasons leading to the triggering of a systemic inflammatory response cascade by increasing the level of arachidonic acid and lipid inflammatory mediators.

In a separate study of a group of patients with severe COVID-19, the markers included cardiolipin 18:0_18:1_22:6_22:6, phosphatidylcholine 16:1_18:2, sphingomyelin d22:0/20:3, and monogalactosyldiacylglycerol 16:0_18:1 (Figure 7). The resulting model was characterized by an accuracy of 84% (CI 28–100%), a sensitivity of 100% (CI 100–100%), and a specificity of 89% (CI 58–100%) at a threshold value of 0.44 (CI 0.01–0.89) (Table 6).

In cases of unsuccessful outcome of IVF, in the group that underwent severe COVID-19, the model showed an increase in the relative concentration of sphingomyelin d22:0/20:3 and cardiolipin 18:0_18:1_22:6_22:6, whereas phosphatidylcholine concentration increased significantly less compared to its sharp increase in models without COVID-19. This observation may indicate the presence of an inflammatory component (from arachidonic acid precursors) in triggering the cascade of reactions prior to miscarriage but not its dominant role in severe COVID-19.

## 3. Discussion

In the present study, a molecular analysis of the FF of patients who underwent mild and severe COVID-19 before entering the IVF program and women who did not have COVID-19 was performed. Several lipids were identified to vary significantly between the groups considered. These lipids belong to the classes of sphingomyelins, cardiolipins, phosphatidylcholines, diacylglycerols, and cholesterol ester. On the basis of the results obtained, models were developed for predicting the threat of miscarriage in patients who had a severe course of COVID-19, along with models for predicting the success of the IVF procedure, depending on the severity of the COVID-19, which have high sensitivity and specificity.

Initially, a model was built to predict the threat of miscarriage for all patients ignoring the COVID-19 status. It included the following lipid markers: lysophosphatidylcholines 16:0 and 18:1, plasmanyl-phosphatidylcholine 16:0/20:4, plasmenyl-phosphatidylethanolamine 16:0/20:4, and cholesterol ester 18:3 (Figure 2). The resulting model had an accuracy of 96% (CI 76–100%), a sensitivity of 95% (CI 65–93%), and a specificity of 93% (CI 59–100%) at a cutoff value of 0.48 (CI 0.01–0.99). The largest upward changes in level were recorded for lysophosphatidylcholine 16:0. Zhang et al. showed a relationship between the age of the patient and an increase in the amount of lysophosphatidylcholine (a precursor of arachidonic acid) in the microenvironment of the follicle. It was suggested that there is a direct relationship between the cascade of inflammatory reactions in the territory of the follicle, caused by oxidative stress and the release of a large number of inflammatory mediators and a decrease in the success of the IVF program in patients of the older age group [20]. Lysophospholipids act as second intracellular messengers or are metabolized into proinflammatory mediators, such as eicosanoids, platelet-activating factors, and lysophosphatidic acid [21]. This cascade of transformations leads to tissue inflammation and impaired hemostasis [22].

Lipids of the phosphatidylcholine class, such as phosphatidylcholine 16:0_16:1, plasmanyl-phosphatidylcholine 16:1/20:4, lysophosphatidylcholine 18:2, and plasmenyl-lysophosphatidylcholine 18:0 (Figure 5), were used to create a model for predicting the success of the IVF program among patients who did not have COVID-19. The resulting model had an accuracy of 81% (CI 54–100%), a sensitivity of 94% (CI 68–100%), and a specificity of 77% (CI 45–100%) at a cutoff value of 0.28 (CI 0.01–0.63). The largest increase in the level was registered in lysophosphatidylcholine 18:2 in cases of unsuccessful IVF outcome. This may indicate that several pathophysiological processes led to the creation of a background that adversely affects the process of embryo implantation and the early stages of pregnancy, leading to termination of the pregnancy. One of the key mechanisms may be the launch of a systemic inflammatory response through the production of arachidonic acid. Arachidonic and linolenic acids are part of the phospholipids of cell membranes from which they are released under the influence of phospholipases. Further transformations of these acids occur via either the cyclooxygenase or the lipoxygenase pathway [23]. Metabolites of arachidonic acid perform important regulatory functions [24,25]. Arachidonic acid derivatives are lipid inflammatory mediators, such as prostaglandins, thromboxanes, and leukotrienes with vaso- and bronchoactive properties. Platelet-activating factor, the most powerful spasmogen, is also formed from membrane phospholipids. This group also includes products of lipid peroxidation—lipoperoxides. In cells and tissues, fatty acids are part of lipids of various classes, are used for the synthesis of steroids, are precursors of prostanoids, and stimulate free oxidation processes, resulting in the formation of peroxidation products and prostaglandins [26,27]. The accumulation of arachidonic acid metabolites in the body provides an inflammatory component. This cascade of reactions is driven by a systemic inflammatory response, and in groups that have had a severe course of COVID-19, it is probably even more catastrophic. Szczuko et al. in their literature review [28] described the role of proinflammatory mediators of arachidonic acid derivatives in pathological conditions associated with reproduction and pregnancy. The authors explained the important role of arachidonic acid derivatives in human fertility and the course of pathological pregnancies. The review presents data from a number of studies demonstrating a strong effect of uncontrolled inflammation on reproduction, spermatogenesis, endometriosis, the genesis of polycystic ovary syndrome (PCOS), implantation, pregnancy, and childbirth. The authors also pointed out that excessive activation of inflammation can lead to miscarriage and other pathological complications during pregnancy [28].

It can be assumed that, under the indirect influence of a viral infection with a severe course of the disease, lipid metabolism in the follicular fluid can cause a change in the oocyte and be closely associated with subsequent pregnancy and delivery. Therefore, at the next stage, models were created to predict the threat of miscarriage and the success of the IVF program in patients who underwent COVID-19. Among patients who underwent mild COVID-19, lipid markers, such as plasmanil-phosphatidylcholine 16:1/18:1, plasmenil-phosphatidylcholine 16:0/18:2, and phosphatidic acid 22:6_22:6 were noted (Figure 6). The resulting model had an accuracy of 88% (CI 60–100%), a sensitivity of 96% (CI 68–100%), and a specificity of 85% (CI 51–100%) at a cutoff value of 0.32 (CI 0.01–0.80). The greatest increase in level in case of an unsuccessful outcome of the IVF program was observed in phosphatidylcholine plasmanil R-16:0/18:2. The predominance of phosphatidylcholines in the model probably indicates an insignificant contribution of mild COVID-19 to pregnancy loss. It can be assumed that, with mild COVID-19, IVF failures are not associated with a past infection and are due to a number of other reasons. The termination of pregnancy is probably triggered by a cascade of systemic inflammatory response, by increasing the level of arachidonic acid and lipid mediators of inflammation. Results of a study by Castiglione Morelli et al. also indicated the absence of COVID-19 and vaccination impact on the outcomes of IVF. However, the authors pointed out two major limitations of their work: a small number of examined women (*n* = 5 recovered from COVID-19 and *n* = 6 vaccinated) and a long time interval between infection or vaccination and oocyte collection [7]. Moreover, the course of COVID-19 was not indicated, which is probably of a fundamental nature. Yaakov Bentov et al. indicated the detection of IgG to SARS-CoV-2 in FF in the absence of impaired steroidogenesis [8]. The authors also concluded that there was no negative impact of COVID-19 on IVF outcomes. However, their study also had several significant limitations, and the main one was a small sample set (*n* = 9 recovered from COVID-19 and *n* = 9 vaccinated), yielding a lack of data on the severity of the disease. Metabolomic analysis of FF was not performed in this study. A study by Yossef Kabalkin et al. confirmed that SARS-CoV-2 does not appear to affect IVF outcomes and early miscarriage rates, despite a slight decrease in sperm concentration in recent recoveries [29].

In the miscarriage prediction model, for patients who underwent severe COVID-19, an increase in lipid markers of other classes was noted. When analyzing the threatened miscarriage model, an increase in the level of cholesterol ester CE 20:5 and monogalactosyldiacylglycerol MGDG 18:0_18:0 was observed (Figure 3). The resulting model had an accuracy of 99% (CI 98–100%), a sensitivity of 99% (CI 98–100%), and a specificity of 99% (CI 98–100%) at a threshold value of 0.50 (CI 0.45–0.56). There was the largest increase in the level of cholesterol ester CE 20:5. Cholesterol esters and fatty acids are the most important components of the body, are necessary for the normal formation of cell membranes, and are of particular importance for brain function. They are also involved in the regulation of cerebral and psychiatric diseases, such as schizophrenia and dementia, including Alzheimer’s disease and multi-infarct dementia.

The model for predicting the success of the IVF program included markers, such as cardiolipin 18:0_18:1_22:6_22:6, phosphatidylcholine 16:1_18:2, sphingomyelin d22:0/20:3, and monogalactosyldiacylglycerol 16:0_18:1 (Figure 7). The resulting model was characterized by an accuracy of 84% (CI 28–100%), sensitivity of 100% (CI 100–100%), and specificity of 89% (CI 58–100%) at a cutoff value of 0.44 (CI 0.01–0.89). In the case of an unsuccessful outcome, the greatest increase in the level of sphingomyelin d22:0/20:3 and cardiolipin 18:0_18:1_22:6_22:6 was noted. Phosphatidylcholine level increased significantly less compared to its sharp increase in non-COVID-19 models. This observation may indicate the presence of an inflammatory component (from arachidonic acid precursors) in triggering the cascade of reactions prior to miscarriage but not its dominant role in severe COVID-19. Sphingomyelin is a type of sphingolipid found in the cell membrane of animals. The myelin sheath of axons of nerve cells is especially rich in this phospholipid. It may be involved in cell signal transduction. Cardiolipin is a phospholipid that is an important component of the inner membrane of mitochondria, and it is necessary for the functioning of numerous enzymes involved in energy metabolism, initiation of apoptosis and tumor growth, and induction of endothelial dysfunction. Baig et al. conducted a study demonstrating differential lipid expression of syncytiotrophoblast microvesicles involved in immune response, coagulation, oxidative stress, and apoptosis in preeclampsia and recurrent miscarriage. As a result of studying the lipid profile of syncytiotrophoblast microvesicles by mass spectrometry in combination with liquid chromatography, the authors quantified approximately 200 lipids, including glycerophospholipids, sphingolipids, free cholesterol, and cholesterol esters (CE) and substantiated their association with recurrent miscarriage [30].

IVF failure after severe COVID-19 is associated with different pathological mechanisms than in the case of a mild infection or in the absence of a history of COVID-19. The endometrium of the uterus and ovarian tissue have been shown to be potential targets for SARS-CoV-2 [31,32]. SARS-CoV-2 infection may subsequently affect ovarian function by altering the molecular and cellular composition of the follicle microenvironment and, thus, affecting oocyte quality in recovered women. In particular, IgG antibodies against SARS-CoV-2 were found in the follicular fluid of IVF patients with a COVID-19 anamneses [22]. In these patients, a change in steroidogenesis in the ovary was also found, with a violation of the migration of endothelial cells. A retrospective study by Michal Youngster et al. also found a significant reduction in pregnancy rates (21% vs. 55%; *p* = 0.006) in patients with SARS-CoV-2 anamneses and cryopreserved embryos (before infection) [33]. This confirms the long-term influence of SARS-CoV-2 on embryo implantation and the need to clarify possible mechanisms.

Abusukhun et al. reported activation of the sphingomyelinase/ceramide pathway in 23 intensive care patients with severe COVID-19. The authors observed an increase in circulating sphingomyelinase activity with subsequent disruption of sphingolipids in serum lipoproteins and in erythrocytes. Consistent with elevated levels of ceramides derived from the inert membrane component sphingomyelin, increased acid sphingomyelinase activity accurately distinguished the intensive care cohort from healthy controls. On the basis of the results, the authors obtained a correlation with biomarkers of a severe clinical phenotype and confirmed the concept of a significant pathophysiological role of acid sphingomyelinase during SARS-CoV-2 infection. The authors concluded that large-scale multicenter trials are currently needed to evaluate the potential benefit of functional inhibition of this sphingomyelinase in critically ill COVID-19 patients [34]. Bruno Silva Andrade, in his publication reviewing clinical conditions and their possible molecular mechanisms in patients with post-COVID complications, concluded that the pathology of COVID-19 is characterized by a cytokine storm that leads to endothelial inflammation, microvascular thrombosis, and multiple organ dysfunction insufficiency [35]. Failed IVF outcomes in patients with a severe course of COVID-19 are likely due to other pathological mechanisms than in the case of a mild course of infection or in the absence of a history of COVID-19. To date, it has been established that the reproductive system (uterine endometrium and ovarian tissue) can be exposed to the SARS-CoV-2 virus [31,32]. Past infection can affect the normal functioning of the ovaries and change the molecular composition of the follicular fluid, thus reducing the quality of oocytes. It has been established that the follicular fluid of patients with COVID-19 contains IgG antibodies against SARS-CoV-2 [36], accompanied by impaired steroidogenesis. This observation confirms the possible delayed impact of the transferred SARS-CoV-2 on embryo implantation and the subsequent course of pregnancy.

Cardiolipin is a potent mediator of endothelial dysfunction involved in the induction of thrombogenesis. This leads to disruption of microcirculation in the fetoplacental complex and, as a result, can lead to pregnancy loss. The endothelium plays an important role in the regulation of the hemostasis system. The transformation of a normal antithrombotic endothelium into a prothrombotic one may be the primary pathophysiological mechanism in an acquired hypercoagulable state against the background of a severe course of a viral disease. The main pathological changes are angiomatosis, microthrombosis, dystrophy of endothelial cells, necrosis and desquamation of endothelial cells, proliferation of intima cells, edema, and plasma impregnation of the basement membrane substance [37]. On the basis of the results obtained, it can be assumed that termination of pregnancy after a severe course of COVID-19 is triggered by a different pathological pathway—the induction of endothelial dysfunction and hypercoagulability. The transformation of the normal antithrombotic status of the endothelium into a prothrombotic status may be the primary pathophysiological mechanism in the acquired hypercoagulable state against the background of a severe course of a viral disease. Further search for the mechanisms of the damaging effect of SARS-CoV-2 on the reproductive system will allow for the development of preventive and therapeutic measures and reducing the number of reproductive losses. This fact may be key in understanding the mechanism of miscarriages and failures of IVF programs after severe COVID-19.

Of practical interest is the possibility of using these predictive models in working with patients who have had COVID-19 before entering the IVF program, since each model has a strictly defined set of lipids, which varies depending on COVID-19 anamneses and the severity of its course. Timely preventive and/or therapeutic actions in patients at risk can improve the outcomes of the IVF program and contribute to a favorable pregnancy outcome.

## 4. Materials and Methods

The work involved samples of follicular fluid collected from 237 women of which 68 had a successful in vitro fertilization procedure. Of these, 12 women miscarried during the first trimester. In total, 103 patients did not have a confirmed SARS-CoV-2 infection, 84 had a mild SARS-CoV-2 infection, and 50 had a severe SARS-CoV-2 infection (Table 7).

Inclusion criteria were as follows: age 18–40 years, normal ovarian reserve (anti-Müllerian hormone (AMH) ≥ 1.2 ng/mL, follicle-stimulating hormone (FSH)) < 12 mIU/mL, and antral follicle count (AF) ≥ 5 in both ovaries), and suffered COVID-19 ≤ 12 months before entering the IVF program for patients in group 2. Exclusion criteria were previous vaccination against COVID-19, contraindications to IVF, morbid obesity (BMI ≥ 40.0 kg/m^2^), donor programs, surrogacy program, HIV infection, polycystic ovary syndrome (PCOS), and endometriosis.

Data on the transferred COVID-19 were obtained from the words of the patients, confirmed by the information entered in the unified state information register, as well as by additional determination of the level of IgG to SARS-CoV-2 in the blood serum above the positivity index (PI). The criterion for a mild form of COVID-19 was subfebrile temperature (<38 °C) in the absence of clinical manifestations of a moderate course of infection. The criteria for the moderate form of COVID-19 included the presence of temperature above 38 °C, shortness of breath during physical exertion, signs of pneumonia with minimal or moderate lung damage (CT 1–2), and the absence of clinical manifestations of a severe course of infection.

To determine antibodies to SARS-CoV-2 in the blood serum, the “Kit of reagents for the detection of class G antibodies to the SARS-CoV-2 spike protein by enzyme immunoassay” (Diagnostic systems; Russia) was used, designed for the qualitative detection of antibodies in serum or plasma of human blood by enzyme immunoassay (ELISA). The result of the analysis was evaluated by the value of the positivity index (PI), calculated as follows: PI = OD of the sample/cutoff, where OD is the optical density. The result was considered positive if the PI value was >1.2, negative if the PI value was <0.8, and doubtful (uncertain) if the PI value was in the range of 0.8–1.2.

Ovarian stimulation was carried out according to the protocol using antagonists of gonadotropin-releasing hormone (ant-GnRH), recombinant follicle-stimulating hormone (rFSH), and/or preparations containing luteinizing hormone (LH). Patients after COVID-19 underwent ovarian stimulation 6 (2–9) months after the disease. At the same time, the dose of gonadotropins was individually selected considering age, anamnesis, and parameters of the ovarian reserve. The introduction of gonadotropins was carried out from the second–third day of the menstrual cycle. Ant-GnRH was administered after the diameter of the leading follicle reached 14 mm daily until the day of the introduction of the ovulation trigger (inclusive), when the diameter of the leading follicle reached 19 mm. Chorionic gonadotropin (CG) (8000–10,000 IU) once or a combination of CG with a gonadotropin-releasing hormone agonist (a-GnRH) was used as an ovulation trigger. Transvaginal puncture (TVP) of the follicles was performed 36 h after the introduction of an ovulation trigger under ultrasound control.

The assessment of the aspirated follicular fluid was carried out by the embryologist using a stereomicroscope. The number of obtained oocyte–cumulus complexes (OCC) was determined, and, after oocyte denudation, their degree of maturity was assessed. In parallel, centrifugation, flotation, and processing of the partner’s sperm were carried out. All mature oocytes were fertilized using the method of IVF or intracytoplasmic sperm injection into the oocyte (ICSI). Normal fertilization was registered by the presence of two symmetrical pronuclei in the cytoplasm 16–18 h after fertilization. After fertilization, the zygotes were transferred to a culture medium (COOK, Australia) for further cultivation. After 120–122 h (on the fifth day) of cultivation, the morphological assessment of the embryos was carried out, considering the morphological characteristics of the embryos according to the Gardner classification: the degree of blastocyst maturity, the quality of the trophectoderm, and the intracellular mass [38].

On the fifth day of cultivation, one or two embryos were transferred into the uterine cavity using a soft catheter in a stimulated cycle. Vaginal micronized progesterone (600 mg daily) or oral dydrogesterone (30 mg daily) was administered to support the post-transfer period.

The onset of pregnancy was determined by the serum level of the beta-subunit of human chorionic gonadotropin (β-hCG) 14 days after embryo transfer (ET) in the uterine cavity. If the level of β-hCG exceeded 20 IU/L, the pregnancy test was considered positive. Then, 21 days after ET, when visualizing the fetal egg in the uterine cavity using ultrasound, a clinical pregnancy was recorded.

Lipids were extracted according to a modified Folch method [39]. Briefly, 480 µL of chloroform/methanol (1/1, *v*/*v*) was added to 40 µL of FF and kept in an ultrasonic bath for 10 min. After that, the mixture was stirred for 10 s and centrifuged for 5 min at 15,000× *g*. The bottom chloroform/methanol layer containing lipids was taken into a separate vial. Then, 250 μL of chloroform/methanol (1/1, *v*/*v*) was added to the aqueous layer and centrifuged for 5 min at 15,000× *g*. The lower chloroform/methanol layer was resampled and combined with the previous portion one. The lipid solution was dried under a stream of nitrogen and redissolved in 200 µL of isopropanol/acetonitrile (1/1, *v*/*v*) for further analysis. The lipid extract was analyzed using high-performance liquid chromatography/mass spectrometry (HPLC–MS) on a Dionex UltiMate 3000 chromatograph (Thermo Scientific, Bremen, Germany) connected to a maXis Impact quadrupole time-of-flight hybrid mass spectrometer (Bruker Daltonics, Bremen, Germany) with the following mass spectrometric analysis parameters: range *m/z* 100–1800, capillary voltage of 4.1 kV in positive ion mode and 3.0 kV in negative ion mode, spray gas pressure of 0.7 bar, drying gas flow rate of 6 L/min, and temperature of 200 °C. Tandem mass spectrometric analysis was performed by data-dependent scan, with a collision energy of 35 eV, an isolation window of 5 Da, and an exclusion time of 2 min. Samples were separated by reverse-phase chromatography on a Zorbax SB-C18 column (150 × 0.5 mm, 3.5 µm, Agilent, Santa Clara, CA, USA) with a gradient from 15% to 45% of eluent B for 2 min and then from 45% to 99% of B for 15 min. As eluent A, a solution of acetonitrile/water (60/40, *v*/*v*) with the addition of 0.1% formic acid and 10 mmol/L of ammonium formate was used. Solvent B was acetonitrile/isopropanol/water (90/8/2 *v*/*v*/*v*) with the addition of 0.1% formic acid and 10 mmol/L ammonium formate. The elution flow rate was 35 µL/min, and the injected sample volume was 0.5 µL.

The raw data obtained during the analysis were converted to the MzXml format using the msConvert software (Proteowizard, 3.0.9987, Palo Alto, CA, USA) and processed with the algorithm provided by Koelmel [40] using MzMine software. Lipid identification was performed using the Lipid Match program by Koelmel [40]. The ion nomenclature was used according to Lipid Maps terminology in abbreviated form [41]. The data were normalized using autoscaling. Statistical analysis was performed in the RStudio environment (1.383 GNU) using scripts in the R language (4.1.1).

Comparative analysis of the lipid profile was carried out using the Mann–Whitney test for pairwise comparison of groups and the Kruskal–Wallis test for simultaneous comparison of groups. The significance threshold was determined to be 0.05. Models for determining the onset of pregnancy and for determining the threat of miscarriage were built on the basis of logistic regression taking into account the interaction between variables. The models had the following form:$$y=\frac{1}{1+e^{-x*\beta^{T}}},$$
where the vector of independent variables x_i_ contains either the value of the i-th marker or the product of one of the other markers or itself (I_i_ or I_i_ × I_j_, or I_i_ × I_i_):

For one marker I,
$$x=\left[I,\;I^{2}\right];$$

For two markers I_1_, I_2_,
$$x=\left[I_{1},\;I_{2},\;I_{1}^{2},\;I_{2}^{2},I_{1}\times I_{2}\right].$$

β^T^ is the transposed vector of coefficients. Construction of x was carried out by direct sequential selection of up to nine markers. The optimal set of markers was selected according to the maximum of the average value of the area under the operating curve of the current model, calculated during cross-validation with 100 cycles and splitting the data with the “training”/“test” ratio of 9/1 [42]. Then, the x vector components were eliminated step by step until the probability of coefficients of each of them being equal to zero was less than 0.05 during cross-validation with 1000 cycles and splitting the data with a “training”/“test” ratio of 9/1. The resulting sensitivity, specificity, and cutoff values were determined as the average values calculated during cross-validation with split data.

## 5. Conclusions

Changes in the lipid composition of the follicular fluid of patients who had mild and severe COVID-19 before entering the IVF program and women who did not have COVID-19 were studied by LC–MS. Several lipids were identified that statistically significantly changed their level, belonging to the classes of sphingomyelins, cardiolipins, phosphatidylcholines, diacylglycerols, and cholesterol esters. Models were developed to predict the threat of miscarriage in patients who had severe COVID-19, along with models to predict the success of the IVF procedure, depending on the severity of COVID-19, with high sensitivity and specificity.

An increase in the level of phosphatidylcholines was noted when analyzing models for predicting the threat of miscarriage and the success of the IVF program for patients who were not ill and had mild COVID-19. This class of lipids is a precursor of arachidonic acid, which, by triggering a systemic inflammatory response, can affect the development of oocytes, the regulation of their maturation, and the early stages of embryogenesis. It can be assumed that, in mild COVID-19, IVF failures are not associated with a past infection and are due to a cascade of systemic inflammatory response reactions, by increasing the level of arachidonic acid and lipid inflammatory mediators.

When analyzing prognosis models among patients who underwent severe COVID-19, there was an increase in lipid markers of other classes: sphingomyelins, cardiolipins, diacylglycerols, and cholesterol ester. The level of phosphatidylcholine in this background increased significantly less compared to its sharp increase in models without COVID-19. This observation may indicate the presence of an inflammatory component (from arachidonic acid precursors) in triggering the cascade of reactions preceding miscarriage but not its dominant role in severe COVID-19.

On the basis of the results obtained, it can be assumed that termination of pregnancy after a severe COVID-19 is triggered by a different pathological pathway, including the induction of endothelial dysfunction and hypercoagulability. The transformation of a normal antithrombotic endothelium into a prothrombotic one may be the primary pathophysiological mechanism in an acquired hypercoagulable state against the background of a severe course of a viral disease. Further search for the mechanisms of the damaging effect of SARS-CoV-2 on the reproductive system will allow for the development of preventive and therapeutic measures and reducing the number of reproductive losses. The revealed changes in the lipid profile of the follicular fluid in patients with severe COVID-19 may serve as therapeutic targets in the future for improving the quality of oocytes and a scientific platform for a new understanding of their microenvironment, including after an inflammatory process.

## Figures and Tables

**Figure 1 ijms-24-00010-f001:**
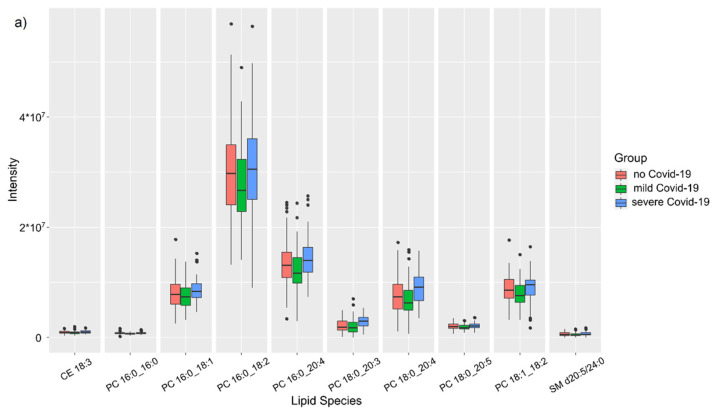
Lipids characterizing differences in the lipid profile of FF between different groups of anamneses of COVID-19: (**a**) in positive ion mode; (**b**) in negative ion mode.

**Figure 2 ijms-24-00010-f002:**
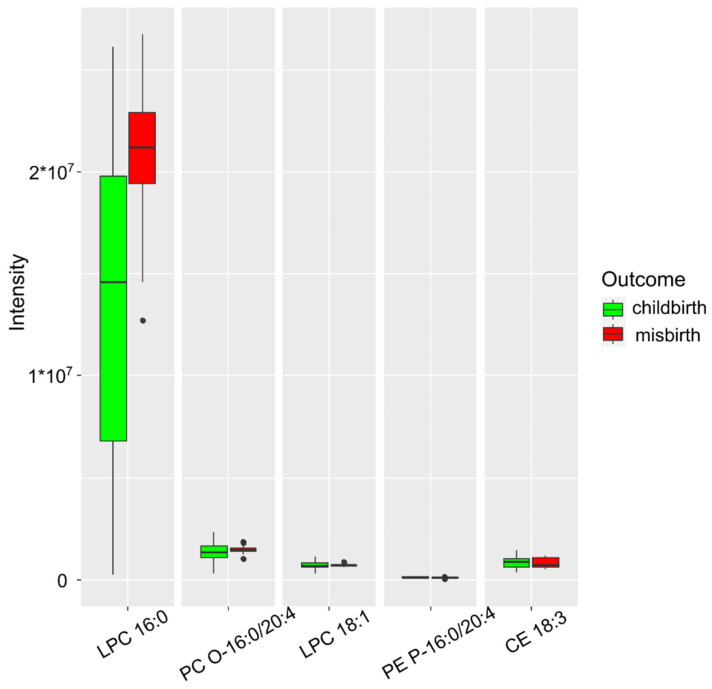
Markers characterizing the presence of a threatened miscarriage, when anamnesis of COVID-19 is ignored.

**Figure 3 ijms-24-00010-f003:**
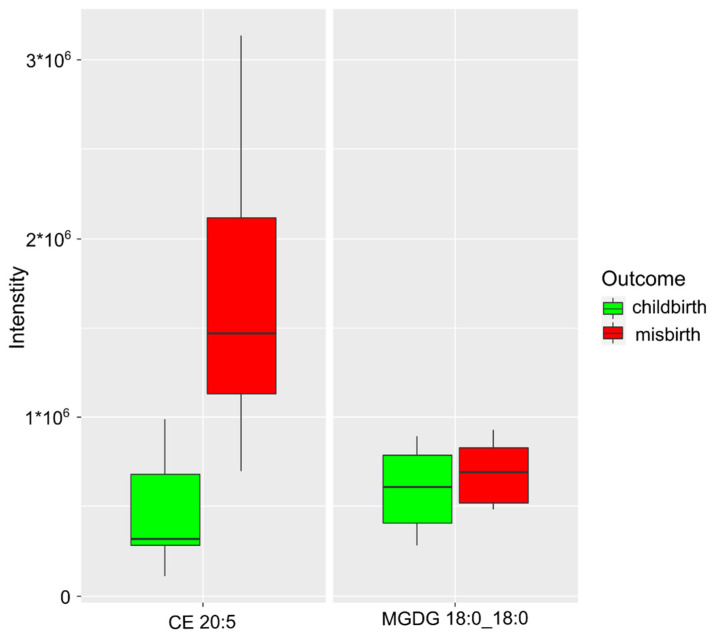
Markers characterizing the threat of miscarriage in those who had a severe form of COVID-19.

**Figure 4 ijms-24-00010-f004:**
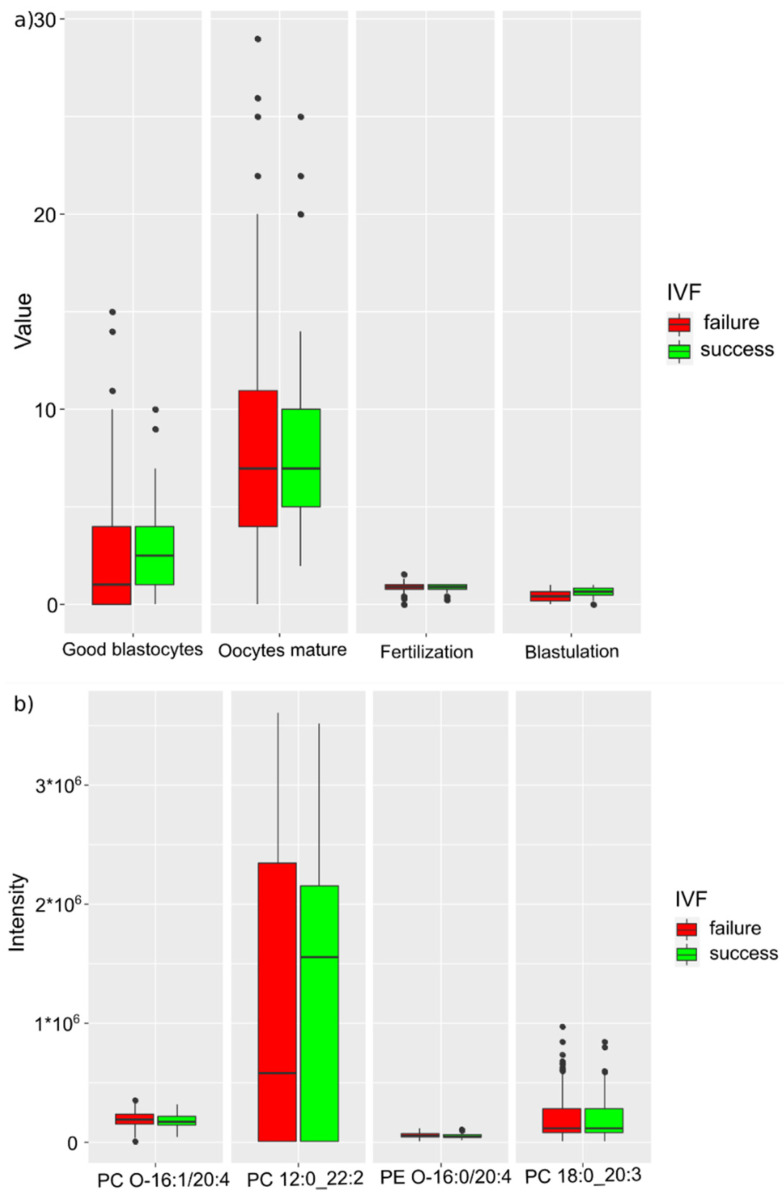
IVF success markers when history of COVID-19 is ignored: (**a**) clinical markers; (**b**) lipids.

**Figure 5 ijms-24-00010-f005:**
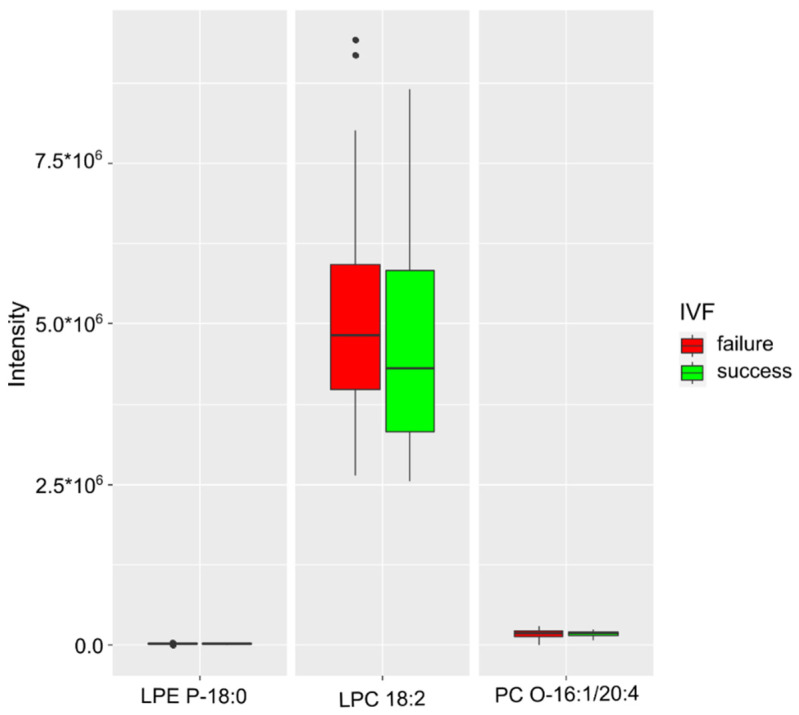
IVF success markers in the COVID-19-free group.

**Figure 6 ijms-24-00010-f006:**
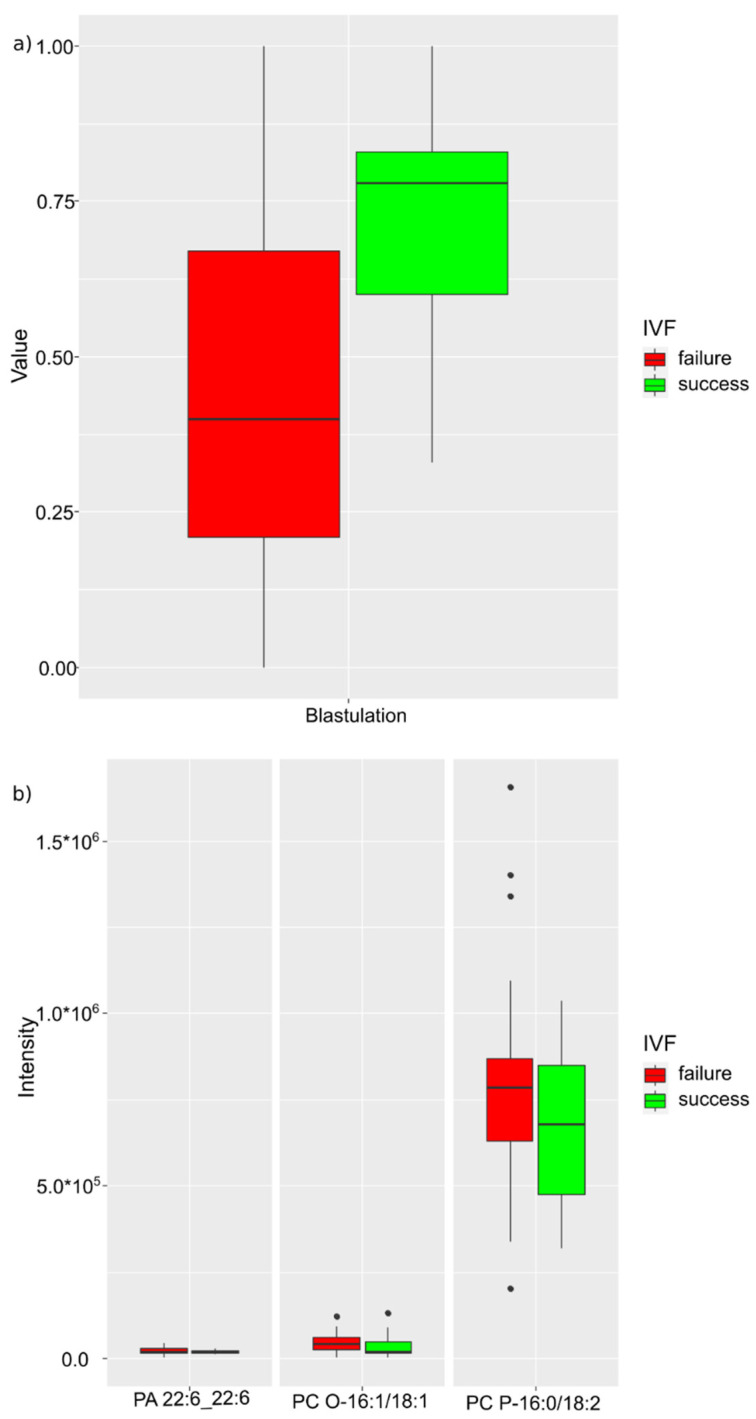
IVF success markers in the group with mild COVID-19: (**a**) clinical markers; (**b**) lipids.

**Figure 7 ijms-24-00010-f007:**
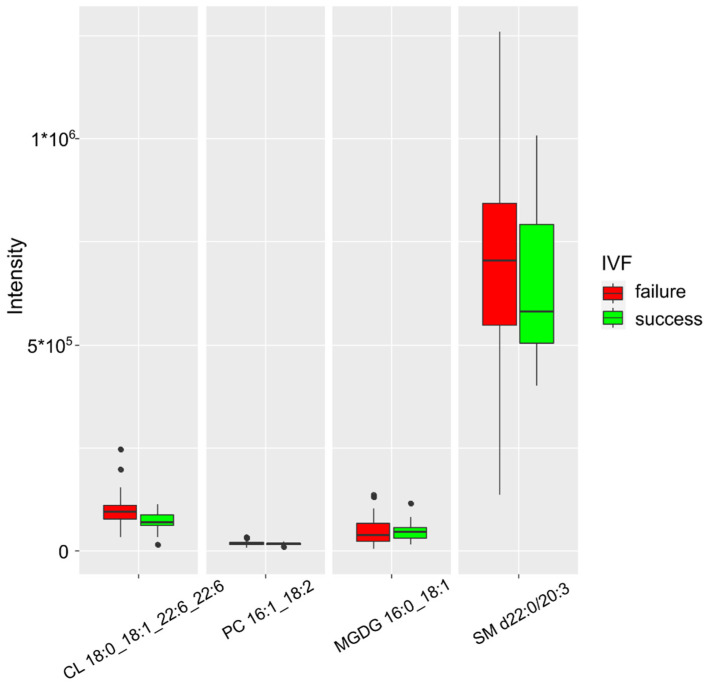
Lipids characterizing IVF success in the group with severe COVID-19.

**Table 1 ijms-24-00010-t001:** Variables of the model used to determine threatened miscarriage when anamnesis of COVID-19 is ignored. (+) positive ion mode; (−) negative ion mode.

Variables	Coefficient (CI)	Wald Criterion	*p*-Value
Intercept	−1.00 × 10^3^ (−1.15 × 10^3^ to −8.49 × 10^2^)	13.24	<0.001
LPC 16:0 (+)	1.33 × 10^−5^ (9.16 × 10^−6^ to 1.74 × 10^−5^)	6.44	<0.001
PC O-16:0/20:4 (+)	5.47 × 10^−4^ (4.37 × 10^−4^ to 6.58 × 10^−4^)	9.90	<0.001
LPC 18:1 (−)	2.40 × 10^−3^ (2.13 × 10^−3^ to 2.67 × 10^−3^)	18.10	<0.001
PE P-16:0/20:4 (−)	−3.53 × 10^−3^ (−5.71 × 10^−3^ to −1.35 × 10^−3^)	−3.24	<0.001
CE 18:3 (+)	−6.33 × 10^−4^ (−8.05 × 10^−4^ to −4.61 × 10^−4^)	−7.35	<0.001
LPC 16:0 (+) × LPC 18:1 (−)	−2.12 × 10^−11^ (−2.74 × 10^−11^ to −1.50 × 10^−11^)	−6.85	<0.001
LPC 16:0 (+) × CE 18:3 (+)	7.39 × 10^−12^ (4.97 × 10^−12^ to 9.81 × 10^−12^)	6.11	<0.001
PC O-16:0/20:4 (+) × LPC 18:1 (−)	2.30 × 10^−10^ (5.06 × 10^−11^ to 4.09 × 10^−10^)	2.56	0.005
PC O-16:0/20:4 (+) × PE P-16:0/20:4 (−)	−3.16 × 10^−9^ (−3.75 × 10^−9^ to −2.56 × 10^−9^)	−10.64	<0.001
LPC 18:1 (−) × CE 18:3 (+)	−7.40 × 10^−10^ (−8.78 × 10^−10^ to −6.02 × 10^−10^)	−10.71	<0.001
PE P-16:0/20:4 (−) × CE 18:3 (+)	3.13 × 10^−9^ (2.49 × 10^−9^ to 4.14 × 10^−9^)	8.03	<0.001
PC O-16:0/20:4 (+) × PC O-16:0/20:4 (+)	−1.04 × 10^−10^ (−1.43 × 10^−10^ to −6.57 × 10^−11^)	−5.41	<0.001
LPC 18:1 (−) × LPC 18:1 (−)	−1.13 × 10^−9^ (−1.26 × 10^−9^ to −9.90 × 10^−10^)	−16.6	<0.001
PE P-16:0/20:4 (−) × PE P-16:0/20:4 (−)	2.03 × 10^−8^ (1.17 × 10^−8^ to 2.90 × 10^−8^)	4.70	<0.001
CE 18:3 (+) × CE 18:3 (+)	3.70 × 10−^10^ (3.13 × 10^−10^ to 4.27 × 10^−10^)	13.00	<0.001

**Table 2 ijms-24-00010-t002:** Variables of the model used to determine the risk of miscarriage for patients with severe COVID-19. (+) positive ion mode; (−) negative ion mode.

Variables	Coefficient (CI)	Wald Criterion	*p*-Value
Intercept	−1.59 × 10^2^ (−1.83 × 10^2^ to −1.35 × 10^2^)	−13.15	<0.001
CE 20:5 (+)	1.00 × 10^−4^ (8.90 × 10^−5^ to 1.11 × 10^−4^)	17.90	<0.001
MGDG 18:0_18:0 (+) × MGDG 18:0_18:0 (+)	1.29 × 10^−10^ (9.87 × 10^−11^ to 1.58 × 10^−10^)	8.61	<0.001

**Table 3 ijms-24-00010-t003:** Variables of the model for determining IVF success without COVID anamneses. (+) positive ion mode; (−) negative ion mode.

Variables	Coefficient (CI)	Wald Criterion	*p*-Value
Intercept	−3.36 × 10^1^ (−4.2 × 10^1^ to −2.5 × 10^1^)	−7.95	<0.001
High-quality blastocysts (HQB)	−6.91 × 10^−1^ (−1.19 to −1.93 × 10^−1^)	−2.78	0.005
Mature oocytes (MO)	8.01 × 10^−1^ (5.53 × 10^−1^ to 1.05)	6.45	<0.001
Fertilization	3.10 × 10^1^ (1.91 × 10^1^ to 4.30 × 10^1^)	5.20	<0.001
Blastulation	1.01 × 10^1^ (5.19 to −1.50 × 10^1^)	4.11	<0.001
PE O-16:0/20:4 (−)	2.54 × 10^−4^ (1.75 × 10^−4^ to 3.33 × 10^−4^)	6.41	<0.001
PC 12:0_22:2 (+)	3.85 × 10^−6^ (2.60 × 10^−6^ to 5.10 × 10^−6^)	6.16	<0.001
PC 18:0_20:3 (−)	1.76 × 10^−5^ (1.05 × 10^−5^ to 2.46 × 10^−5^)	4.96	<0.001
PC O-16:1/20:4 (−)	4.36 × 10^−5^ (1.55 × 10^−5^ to 7.16 × 10^−5^)	3.11	0.002
«HQB» × «MO»	8.82 × 10^−2^ (4.83 × 10^−2^ to 1.28 × 10^−1^)	4.42	<0.001
«HQB» × PE O-16:0/20:4 (−)	1.07 × 10^−5^ (2.94 × 10^−6^ to 1.85 × 10^−5^)	2.76	0.006
«HQB» × PC 12:0_22:2 (+)	1.94 × 10^−7^ (9.02 × 10^−8^ to 2.97 × 10^−7^)	3.74	<0.001
«MO» × PE O-16:0/20:4 (−)	−1.00 × 10^−5^ (−1.35 × 10^−5^ to −6.61 × 10^−6^)	−5.84	<0.001
«MO» × PC 12:0_22:2 (+)	−1.67 × 10^−7^ (−2.17 × 10^−7^ to −1.7 × 10^−7^)	−6.66	<0.001
«Fertilization» × «blastulation»	−8.81 (−1.28 × 10^1^ to −4.78)	−4.38	<0.001
«Fertilization» × PE O-16:0/20:4 (−)	−8.52 × 10^−5^ (−1.49 × 10^−4^ to −2.09 × 10^−5^)	−2.65	0.008
«Fertilization» × PC 12:0_22:2 (+)	−1.39 × 10^−6^ (−2.37 × 10^−6^ to −4.07 × 10^−7^)	−2.83	0.005
«Fertilization» × PC 18:0_20:3 (−)	−1.50 × 10^−5^ (−2.20 × 10^−5^ to −8.09 × 10^−6^)	−4.33	<0.001
«Fertilization» × PC O-16:1/20:4 (−)	−4.21 × 10^−5^ (−6.96 × 10^−5^ to −1.46 × 10^−5^)	−3.06	0.002
«Blastulation» × PE O-16:0/20:4 (−)	−5.33 × 10^−5^ (−9.32 × 10^−5^ to −1.34 × 10^−5^)	−2.67	0.007
«Blastulation» × PC O-16:1/20:4 (−)	1.97 × 10^−5^ (3.54 × 10^−6^ to 3.58 × 10^−5^)	2.44	0.01
PE O-16:0/20:4 (−) × PC 12:0_22:2 (+)	−1.13 × 10^−12^ (−1.90 × 10^−11^ to −3.62 × 10^−12^)	−2.94	0.003
PE O-16:0/20:4 (−) × PC 18:0_20:3 (−)	−8.63 × 10^−11^ (−1.26 × 10^−10^ to −4.63^−11^)	−4.31	<0.001
PC 12:0_22:2 (+) × PC 18:0_20:3 (−)	9.51 × 10^−13^ (2.123 × 10^−13^ to 1.69 × 10^−12^)	2.57	0.01
«HQB» × «HQB»	−1.31 × 10^−1^ (−1.83 × 10^−1^ to −7.77 × 10^−2^)	−4.94	<0.001
«MO» × «MO»	−2.09 × 10^−2^ (−3.25 × 10^−2^ to −9.26 × 10^−3^)	−3.59	<0.001
«Fertilization» × «fertilization»	−5.35 (−1.01 × 10^1^ to −5.89 × 10^−1^)	−2.45	0.02
PE O-16:0/20:4 (−) × PE O-16:0/20:4 (−)	−7.01 × 10^−10^ (−9.81 × 10^−10^ to −4.21 × 10^−10^)	−5.01	<0.001
PC 12:0_22:2 (+) × PC 12:0_22:2 (+)	−5.30 × 10^−13^ (−7.58 × 10^−13^ to −3.03 × 10^−13^)	−4.66	<0.001
PC O-16:1/20:4 (−) × PC O-16:1/20:4 (−)	−5.00 × 10^−11^ (−9.23 × 10^−11^ to −7.85 × 10^−12^)	−2.37	0.02

**Table 4 ijms-24-00010-t004:** Model variables used to determine IVF success for the COVID-19 free group. (+) positive ion mode; (−) negative ion mode.

Variables	Coefficient (CI)	Wald Criterion	*p*-Value
Intercept	2.98 × 10^1^ (2.12 × 10^1^ to 3.85 × 10^1^)	6.88	<0.001
PC 16:0_16:1 (+)	−8.53 × 10^−5^ (−1.13 × 10^−4^ to −5.78 × 10^−5^)	−6.20	<0.001
LPE P-18:0 (−)	−5.47 × 10^−4^ (−7.45 × 10^−4^ to −3.49 × 10^−4^)	−5.52	<0.001
LPC 18:2 (+)	−2.27 × 10^−6^ (−3.61 × 10^−6^ to −9.35 × 10^−7^)	−3.40	<0.001
PC 16:0_16:1 (+) × LPE P-18:0 (−)	1.98 × 10^−9^ (1.14 × 10^−9^ to 2.82 × 10^−9^)	4.72	<0.001
PC 16:0_16:1 (+) × PC O-16:1/20:4 (−)	1.50 × 10^−10^ (7.11 × 10^−11^ to 2.89 × 10^−10^)	3.80	<0.001
PC 16:0_16:1 (+) × LPC 18:2 (−)	8.31 × 10^−12^ (4.35 × 10^−12^ to 1.23 × 10^−11^)	4.19	<0.001
LPE P-18:0 (−) × PC O-16:1/20:4 (−)	1.30 × 10^−9^ (4.10 × 10^−10^ to 2.17 × 10^−9^)	2.93	0.002
LPE P-18:0 (−) × LPC 18:2 (−)	−8.07 × 10^−11^ (−1.24 × 10^−10^ to −3.79 × 10^−11^)	−3.77	<0.001
PC O-16:1/20:4 (−) × LPC 18:2 (−)	6.17 × 10^−12^ (2.10 × 10^−12^ to 1.02 × 10^−11^)	3.03	0.001
PC 16:0_16:1 (+) × PC 16:0_16:1 (+)	−5.69 × 10^−11^ (−1.09 × 10^−10^ to −4.65 × 10^−12^)	−2.18	0.02
PC O-16:1/20:4 (−) × PC O-16:1/20:4 (−)	−3.27 × 10^−10^ (−4.14 × 10^−10^ to −2.39 × 10^−10^)	−7.45	<0.001

**Table 5 ijms-24-00010-t005:** Model variables used to determine IVF success in mild COVID-19 group. (+) positive ion mode; (−) negative ion mode.

Variables	Coefficient (CI)	Wald Criterion	*p*-Value
Intercept	−6.64 (−1.16 × 10^1^ to −1.72)	−2.70	0.003
PA 22:6_22:6 (−)	7.73 × 10^−4^ (2.08 × 10^−4^ to 1.34 × 10^−3^)	2.74	0.003
Blastulation × PC O-16:1/18:1 (−)	−5.49 × 10^−4^ (−7.45 × 10^−4^ to −3.53 × 10^−4^)	−5.61	<0.001
Blastulation × PC P-16:0/18:2 (+)	6.96 × 10^−5^ (4.29 × 10^−5^ to 9.64 × 10^−5^)	5.20	<0.001
PC O-16:1/18:1 (−) × PA 22:6_22:6 (−)	−8.60 × 10^−9^ (−1.43 × 10^−8^ to −2.89 × 10^−9^)	−3.01	0.001
PC O-16:1/18:1 (−) × PC P-16:0/18:2 (+)	7.10 × 10^−10^ (4.27 × 10^−9^ to 9.94 × 10^−10^)	5.02	<0.001
Blastulation × blastulation	−1.56 × 10^1^ (−2.36 × 10^1^ to −7.60)	−3.90	<0.001
PA 22:6_22:6 (−) × PA 22:6_22:6 (−)	−1.30 × 10^−8^ (−2.30 × 10^−8^ to −2.93 × 10^−9^)	−2.58	0.004
PC P-16:0/18:2 (+) × PC P-16:0/18:2 (+)	−5.66 × 10^−11^ (−7.86 × 10^−11^ to −3.46 × 10^−11^)	−5.14	<0.001

**Table 6 ijms-24-00010-t006:** Model variables used to determine IVF success in the severe COVID-19 survivor group. (+) positive ion mode; (−) negative ion mode.

Variables	Coefficient (CI)	Wald Criterion	*p*-Value
Intercept	4.89 (3.58 to 6.20)	7.47	<0.001
CL 18:0_18:1_22:6_22:6 (−)	−5.97 × 10^−5^ (−7.64 × 10^−5^ to −4.3 × 10^−5^)	−7.15	<0.001
PC 16:1_18:2 (−) × SM (d22:0_20:3) (−)	−3.12 × 10^−10^ (−3.99 × 10^−10^ to −2.25 × 10^−10^)	−7.16	<0.001
MGDG 16:1_18:2 (−) × SM (d22:0_20:3) (−)	2.49 × 10^−10^ (1.88 × 10^−10^ to 3.09 × 10^−10^)	8.29	<0.001
MGDG 16:1_18:2 (−) × MGDG 16:1_18:2 (−)	−1.54 (−1.92 × 10^−9^ to −1.17 × 10^−9^)	−8.20	<0.001

**Table 7 ijms-24-00010-t007:** Distribution of IVF results across groups with differences in history of SARS-CoV-2 (number of patients).

	No SARS-CoV-2 (*n* = 103)	Mild SARS-CoV-2 (*n* = 84)	Severe SARS-CoV-2 (*n* = 50)
No pregnancy	73 (70.9%)	63 (75%)	33 (66%)
Pregnancy, childbirth	27 (26.2%)	18 (21.4%)	11 (22%)
Pregnancy, miscarriage	3 (2.9%)	3 (3.6%)	6 (12%)

## Data Availability

Not applicable.

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
