# Peer review of "Past COVID-19: The Impact on IVF Outcomes Based on Follicular Fluid Lipid Profile"

_ijms, 2022, doi:10.3390/ijms24010010_

Round 1

Reviewer 1 Report

The manuscript entitled “Past COVID-19: the impact on IVF outcomes based on follicular fluid lipid profile” by Lomova et al., reports interesting results on the impact of COVID-19 infection on the follicular fluid characteristics and functions. By applying a mass spectrometry approach, Authors identified the lipid profiles in follicular fluids of women who underwent COVID-19 in mild and severe forms before entering the IVF program. Notwithstanding the manuscript is reporting interesting data that could shed light on the impact of COVID-19 on female fertility, some minor revisions are kindly requested:

1) The introduction appears too brief and does not report sufficient information on the possible impact of COVID-19 on follicular fluid composition. Moreover, Authors are kindly invited to better explain more in details reasons why they focused on the lipid profiles in follicular fluids.

2) Results are not clearly reported in the specific section. In particular, Authors are kindly invited to better report and explain the different lipid profiles in the various patient groups. In light of the above, it is unclear from the manuscript, based on the results section, the real contribution of COVID-19 infection on IVF outcomes.

3) In the discussion section, Authors are kindly invited to better explain the significance of the lipids identified as possible biomarkers to monitor and predict IVF outcomes in women affected by COVID-19 infection, and their involvement in biological processes related to viral infection and/or imbalance of follicular fluid lipid composition.

4) In the Materials and Methods section, Authors are kindly invited to better describe the study population, focusing the attention on the composition of the different groups of patients.

Author Response

Reviewer 1

The manuscript entitled “Past COVID-19: the impact on IVF outcomes based on follicular fluid lipid profile” by Lomova et al., reports interesting results on the impact of COVID-19 infection on the follicular fluid characteristics and functions. By applying a mass spectrometry approach, Authors identified the lipid profiles in follicular fluids of women who underwent COVID-19 in mild and severe forms before entering the IVF program. Notwithstanding the manuscript is reporting interesting data that could shed light on the impact of COVID-19 on female fertility, some minor revisions are kindly requested:

1) The introduction appears too brief and does not report sufficient information on the possible impact of COVID-19 on follicular fluid composition. Moreover, Authors are kindly invited to better explain more in details reasons why they focused on the lipid profiles in follicular fluids

Answer:

To clarify the reasons why we focused on lipid profiles in FF, a reference was added to a study by Montani, D.A. et al. , to discover new links in the pathogenesis of a number of diseases leading to infertility. (Montani, D. A., Braga, D. P. A. F., Borges, E., Jr, Camargo, M., Cordeiro, F. B., Pilau, E. J., … Lo Turco, E. G. (2019). Understanding mechanisms of oocyte development by follicular fluid lipidomics. Journal of Assisted Reproduction and Genetics, 36(5), 1003–1011.)”.

To date, experience has been accumulated in the management of patients with COVID-19, however, the molecular mechanisms of the pathogenesis of the disease and its indirect effect on various organs and systems have been little studied. We hypothesized in the manuscript that “It is known that patients with severe COVID-19 are exposed to a “cytokine storm” on the body, which causes a systemic inflammatory response and can damage any organs and systems, including the reproductive one [16,17]. Therefore, changes in the plasma composition can also be reflected in the FF composition. In addition, several biologically active substances released during inflammation modulate lipid metabolism [18], which indicates a possible relationship between the severity of COVID-19, the detection of certain metabolites, and the outcome of IVF” (page 2 in the manuscript).

2) Results are not clearly reported in the specific section. In particular, Authors are kindly invited to better report and explain the different lipid profiles in the various patient groups. In light of the above, it is unclear from the manuscript, based on the results section, the real contribution of COVID-19 infection on IVF outcomes

Answer:

For a deeper understanding of the data obtained, literature data has been added to the Results section, explaining the difference in lipid profiles in different groups of patients and supplementing the reader's understanding of the contribution of COVID-19 infection to IVF outcomes, namely:

“Of interest is the increase in the 16:0 concentration of lysophosphatidylcholine in the model. Lysophospholipids are precursors of arachidonic acid and act as second intracellular messengers. This cascade of transformations leads to tissue inflammation and im-paired hemostasis. Szczuko M. et al. in his literature review[19] describe the role of pro-inflammatory mediators of arachidonic acid derivatives in pathological conditions associated with reproduction and pregnancy. The authors explain the important role of arachidonic acid derivatives in human fertility and the course of pathological pregnancies. The review presents data from a number of studies demonstrating a strong effect of uncontrolled inflammation on reproduction, spermatogenesis, endometriosis, the genesis of polycystic ovary syndrome (PCOS), implantation, pregnancy and childbirth. The au-thors also point out that excessive activation of inflammation can lead to miscarriage and other pathological complications during pregnancy [19]”

 Also added the following text:

“It should be noted that in the resulting miscarriage prediction model, there was a significant increase in the relative concentration of cholesterol ester CE 20:5. Nicotra M. et al. showed that free fatty acid levels were significantly higher in women with recurrent abortion. The authors point out that an increase in the concentration of these molecules in women with recurrent miscarriage probably led to a stress-dependent release of catecholamines into the blood, followed by uterine vasoconstriction and placental ischemia. These mechanisms, combined with additional damage caused by reoxygenation, led to miscarriage [20]. Baig S. et al. conducted a study demonstrating differential lipid expression of syncytiotrophoblast microvesicles involved in immune response, coagulation, oxidative stress and apoptosis in preeclampsia and recurrent miscarriage. As a result of studying the lipid profile of syncytiotrophoblast microvesicles by mass spectrometry in combination with liquid chromatography, the authors quantified about 200 lipids, including glycerophospholipids, sphingolipids, free cholesterol, and cholesterol esters (CE) and substantiated their association with recurrent miscarriage [21]”.

“Lipids of the phosphatidylcholine class, such as phosphatidylcholine 16:0_16:1, plasmanyl-phosphatidylcholine 16:1/20:4, lysophosphatidylcholine 18:2, and plasmenyl-lysophosphatidylcholine 18:0 (Figure 5) were used to create a model for predicting the success of the IVF program among patients who did not have COVID-19. The resulting model has an accuracy of 81% (CI 54% - 100%), a sensitivity of 94% (CI 68% - 100%), and a specificity of 77% (CI 45% - 100%) at a cut-off value of 0.28 (CI 0.01-0.63). The largest in-crease in the level was registered in lysophosphatidylcholine 18:2 in cases of unsuccessful IVF outcome. This may indicate that several pathophysiological processes lead to the creation of a background that adversely affects the process of embryo implantation and the early stages of pregnancy, leading to termination of pregnancy. One of the key mechanisms may be the launch of a systemic inflammatory response, through the pro-duction of arachidonic acid.”

 “In patients in cases of failure of IVF programs who underwent mild COVID-19, an increase in the level of plasmenil phosphatidylcholine P-16:0/18:2 was registered. It can be assumed that IVF failures in mild cases of COVID-19 are not directly related to the past infection and are probably due to other reasons leading to the triggering of a systemic inflammatory response cascade by increasing the level of arachidonic acid and lipid inflammatory mediators”.

“In cases of unsuccessful outcome of IVF, in the group who underwent severe COVID-19, the model showed an increase in the relative concentration of sphingomyelin d22:0/20:3 and cardiolipin 18:0_18:1_22:6_22:6. Whereas phosphatidylcholine concentration in-creased significantly less compared to its sharp increase in models without COVID-19. This observation may indicate the presence of an inflammatory component (from arachidonic acid precursors) in triggering the cascade of reactions prior to miscarriage, but not its dominant role in severe COVID-19. Abusukhun M. et al. reported activation of the sphingomyelinase-ceramide pathway in 23 intensive care patients with severe COVID-19. The authors observed an increase in circulating sphingomyelinase activity with subsequent disruption of sphingolipids in serum lipoproteins and in erythrocytes. Consistent with elevated levels of ceramides derived from the inert membrane component sphingomyelin, increased acid sphingomyelinase activity accurately distinguished the intensive care cohort from healthy controls. Based on the results, the authors obtained a correlation with biomarkers of severe clinical phenotype and confirm the concept of a significant pathophysiological role of acid sphingomyelinase during SARS-CoV-2 infection. The authors concluded that large-scale multicenter trials are currently needed to evaluate the potential benefit of functional inhibition of this sphingomyelinase in critically ill COVID-19 patients [22]. Bruno Silva Andrade, in his publication reviewing clinical conditions and their possible molecular mechanisms in patients with post-COVID complications. He concludes that the pathology of COVID-19 is characterized by a cytokine storm that leads to endothelial inflammation, microvascular thrombosis, and multiple organ dysfunction insufficiency [23]. Failed IVF outcomes in patients with a severe course of COVID-19 are likely due to other pathological mechanisms than in the case of a mild course of infection or in the absence of a history of COVID-19. To date, it has been established that the reproductive system (uterine endometrium and ovarian tissue) can be exposed to the SARS-CoV-2 virus [24,25]. Past infection can affect the normal functioning of the ovaries, change the molecular composition of the follicular fluid and thus reduce the quality of oocytes. It has been established that the follicular fluid of patients with COVID-19 in history contains IgG antibodies against SARS-CoV-2 [26], accompanied by impaired steroidogenesis. This observation confirms the possible delayed impact of the transferred SARS-CoV-2 on embryo implantation and the subsequent course of pregnancy”.

3) In the discussion section, Authors are kindly invited to better explain the significance of the lipids identified as possible biomarkers to monitor and predict IVF outcomes in women affected by COVID-19 infection, and their involvement in biological processes related to viral infection and/or imbalance of follicular fluid lipid composition.

Answer:

The discussion section is devoted to the role of lipids identified as possible biomarkers, namely: phosphatidylcholines 16:0_16:1, plasmanyl-O-16:1/20:4, lysophosphatidylcholine 18:2, lysophosphatidylethanolamine P-18:0, phosphatidylcholines plasmanyl O- 16:1/18:1 and Plasmenil P-16:0/18:2, phosphatidylic acid 22:6_22:6, cholesterol ester CE 20:5, monogalactosyldiacylglycerol MGDG 18:0_18:0, cardiolipin 18:0_18:1_22:6_22 :6, sphingomyelin d22:0/20:3 and monogalactosyldiacylglycerol 16:0_18:1) in the possible perspective of monitoring and predicting the results of IVF in women affected by COVID-19 infection and their participation in biological processes.

Considering the wishes of other two reviewers (for example: “Discussion is interesting but it does not need such an extensive pathophysiological description at several lines as they are widely known. I would suggest the authors to reduce some lines of literature review in the discussion”), we did not increase the discussion section for by adding even more detailed explanations.

4) In the Materials and Methods section, Authors are kindly invited to better describe the study population, focusing the attention on the composition of the different groups of patients.

Answer:

For a more in-depth description of the study population, the description of inclusion and exclusion criteria was expanded.

Reviewer 2 Report

This is my review on the Article

“Past COVID-19: the impact on IVF outcomes based on follicular fluid lipid profile”. 

The authors aim to evaluate through mass spectrometry the changes on the follicular fluid of patients who underwent Covid-19. Abstract is very interesting but at some point, it feels like “discussion”. I would suggest authors to review their abstract into a stricter presentation of results and conclusions. 

Introduction and aim are fine. 

Results are interestingly presented. My suggestion is to minimize the size of the figures. Material and methods are adequately described. 

Discussion is interesting but it does not need such an extensive pathophysiological description at several lines as they are widely known. I would suggest the authors to reduce some lines of literature review in the discussion. 

Conclusion section must be revised. Several lines should be moved to the introduction or discussion section. Authors should make an effort to present in 7 to 10 lines the interpretation of their findings. 

Author Response

Reviewer 2

The authors aim to evaluate through mass spectrometry the changes on the follicular fluid of patients who underwent Covid-19. Abstract is very interesting but at some point, it feels like “discussion”. I would suggest authors to review their abstract into a stricter presentation of results and conclusions.

Answer:

The abstract has been completely revised:

“Abstract: Follicular fluid is an important component of follicle growth and development. Negative effects of COVID-19 on follicular function are still open. The aim of this work is to study the features of the lipid profile of the follicular fluid and evaluate the results of the In Vitro Fertilization (IVF) program in women after COVID-19 to identify biomarkers with prognostic potential. The work involved samples of follicular fluid collected from 237 women. Changes in the lipid composition of the follicular fluid of patients who underwent COVID-19 in mild and severe forms before entering the IVF program, and women who did not have COVID-19 were studied by mass spectrometry. Several lipids have been identified that significantly change their level. Based on these findings, models were developed for predicting the threat of miscarriage in patients who had a severe course of COVID-19 and models for predicting the success of the IVF procedure, depending on the severity of COVID-19. Of practical interest is the possibility of using the developed predictive models in working with patients who have undergone COVID-19 before entering the IVF program. The re-sults of the study suggest that the onset of pregnancy and its outcome after severe COVID-19 may be associated with changes in lipid metabolism in the follicular fluid”.

Results are interestingly presented. My suggestion is to minimize the size of the figures. Material and methods are adequately described.

Answer:

The size of the figures has been reduced.

Discussion is interesting but it does not need such an extensive pathophysiological description at several lines as they are widely known. I would suggest the authors to reduce some lines of literature review in the discussion

Answer:

In view of another reviewer's wish to expand the Discussion section, we encountered difficulties in making a decision and did not shorten this section.

Conclusion section must be revised. Several lines should be moved to the introduction or discussion section. Authors should make an effort to present in 7 to 10 lines the interpretation of their findings.

Answer:

The conclusion section has been revised and shortened.

Reviewer 3 Report

In the present study the authors developed some models to predict either the threat of miscarriage or the potential success of the IVF procedures in patients who had COVID-19 at different severities (low, mild, and severe). By using mass spectrometry on patients’ follicular fluids, they found that the concentration of several lipids was significantly changed. This is a potentially interesting study, but to be considered simply as a preliminary as important parameters are completely missing . The whole procedure of IVF is seriously affected by any inflammatory status of both partners. This is an important issue that must be more seriously addressed in this study. The mere evaluation of COVID-Not COVID status is insufficient to explain the results (if any) of the study

Major issues to be addressed for the paper to be considered

·         Abstract: The authors state that “It can be assumed that in the mild COVID-19, IVF failures are not associated with a past infection and are due to a cascade of systemic inflammatory response reactions” and “the induction of endothelial dysfunction and hypercoagulablity…etc”. This is a mere assumption and should be considered in short and at the end since the authors didn’t show any data about.

·         Methods: For the IVF, patients’ evaluation should be elucidated. Which was the diagnosis for these women? Were They PCOS? Endometriosis? Hypertensive? And how were their partners diagnosed? This part of the anamnesis is very important as it clarifies not only the formation of the embryos, but also their attachment and development. Moreover, the whole practice of IVF is seriously affected by any inflammatory status of both partners. This is an important issue that must be more seriously addressed in this study. The mere evaluation of COVID-Not COVID status is insufficient.

The validation of their predictive models needs to be confirmed, for example by analyzing each patient’s data into a grid showing the different COVID disease severity and assessing their match

Author Response

Reviewer 3

In the present study the authors developed some models to predict either the threat of miscarriage or the potential success of the IVF procedures in patients who had COVID-19 at different severities (low, mild, and severe). By using mass spectrometry on patients’ follicular fluids, they found that the concentration of several lipids was significantly changed. This is a potentially interesting study, but to be considered simply as a preliminary as important parameters are completely missing. The whole procedure of IVF is seriously affected by any inflammatory status of both partners. This is an important issue that must be more seriously addressed in this study. The mere evaluation of COVID-Not COVID status is insufficient to explain the results (if any) of the study

Answer:

To date, two main types of metabolomics studies are carried out by the MS method: non-targeted or targeted. The first, non-targeted, allows the simultaneous evaluation of thousands of metabolites in a sample. In off-target analysis, it is possible to determine not the absolute amounts of compounds, but the relative change between groups. They are then usually applied to hypotheses in an attempt to understand biological processes. Subsequently, they should be tested in large-scale studies. Our study can be classified as non-targeted.

Major issues to be addressed for the paper to be considered

Abstract: The authors state that “It can be assumed that in the mild COVID-19, IVF failures are not associated with a past infection and are due to a cascade of systemic inflammatory response reactions” and “the induction of endothelial dysfunction and hypercoagulablity…etc”. This is a mere assumption and should be considered in short and at the end since the authors didn’t show any data about.

Answer:

The abstract has been completely revised.

Methods: For the IVF, patients’ evaluation should be elucidated. Which was the diagnosis for these women? Were They PCOS? Endometriosis? Hypertensive? And how were their partners diagnosed? This part of the anamnesis is very important as it clarifies not only the formation of the embryos, but also their attachment and development. Moreover, the whole practice of IVF is seriously affected by any inflammatory status of both partners. This is an important issue that must be more seriously addressed in this study. The mere evaluation of COVID-Not COVID status is insufficient.

Answer:

For a more in-depth description of the study population, in the Materials and Methods section, the description of the inclusion and exclusion criteria in the groups has been expanded:

As clinicians, we are well aware that PCOS and endometriosis cause infertility and can alter the lipid composition of the blood and follicular fluid. Naturally, we did not take such patients into the study. We included them in the exclusion criteria:

«Inclusion criteria were age 18-40 years, normal ovarian reserve (anti-Müllerian hormone (AMH) ≥1.2 ng/mL, follicle-stimulating hormone (FSH)) <12 mIU/mL, antral follicle count (AF) ≥5 in both ovaries), suffered COVID-19 12 months or less before en-tering the IVF program for patients in group 2. Exclusion criteria were vaccination against COVID-19 in history, contraindications to IVF, morbid obesity (BMI ≥40.0 kg/m2), donor programs, surrogacy program, HIV infection, polycystic ovary syndrome (PCOS), and endometriosis».

The validation of their predictive models needs to be confirmed, for example by analyzing each patient’s data into a grid showing the different COVID disease severity and assessing their match

Answer:

The purpose of our work was to study the features of the lipid profile of the follicular fluid and evaluate the results of the IVF program in women undergoing IVF treatment after suffering COVID-19 in order to identify biomarkers with prognostic potential. Based on your valuable comments and suggestions, we will try to provide a deep clinical analysis of patients and the subsequent adaptation of prognostic models in the routine practice in subsequent publications, in which we will be able to focus the reader's attention on these aspects of the study.

Reviewer 4 Report

In this study the authors aimed at exploring the impact of COVID-19 infection on the lipid profiles of the follicular fluid to identify biomarkers with prognostic potential. They found significant differences in the follicular fluid concentrations of several lipids when non-Covid patients were compared with Covid patients that developed moderate and severe symptoms. To the best of my knowledge the reported results are novel, interesting and worth publishing. However, the ms in its present form suffers from four major limitations that should be addressed by the authors in case the handling editor invites resubmission of the suitably revised version of the ms.

1. Lack of mechanistic focus: The paper is very descriptive and lacks mechanistic focus. The authors quantified a number of randomly selected readout parameters and performed statistical correlation analyses. They found statistically significant differences between the different experimental groups but the mechanistic basis for these differences has not been explored. The authors suggest severe Covid-19 infections induce endothelial dysfunction and hypercoagulability (this has been reported many times before by other researchers) and that these changes might impact the lipid composition of the follicular fluid. However, the authors do not explore at the molecular level how exactly systemic endothelial dysfunction and hypercoagulability (pro-thrombotic status) induces the observed alterations in the lipid composition of the follicular fluid.

2. The ms is very difficult to read and I experienced problems when I attempted to extract and to evaluate the major results of this study. The Abstract is far to long (362 words) and does not present the major experimental findings in a concise way. Instead, it involves a number of general statements and speculations, which are not really supported by the experimental data. This is also the case for the Results and Discussion section. Except for the Mat+Meth section all other parts of the ms should be restructured and the authors should focus the attention of the readers on their experimental findings.

3. Inadequate data presentation and biological relevance of the observed differences: When I look at the Fig. 1a+b I do not see major differences between the experimental groups for any of the determined readout parameter. The differences might be statistically significant but the extent of these differences is really subtle. Thus, for me it is questionable whether these differences are biologically relevant and whether or not these measures can be considered predictive readout parameters. The same is true for Fig. 2. Here again, the extent of the difference between the childbirth and the miscarriage group is very low for most of the readout parameters. Only for LPC-16:0 a difference is easily appreciated but even for this readout parameter the difference between the means is not even a factor of 2. This is also the case for the results presented in Figure 3-7. In general, statistic significance does not automatically mean that these differences are of biological relevance.

4. Retrospective study: I am not an epidemiologist but I am a little worried about the value of such retrospective studies. Reading the Met+Meth section I learned that analysis of the lipid composition of the follicular fluid was not blinded. Moreover, assignment of the patients to the two different Covid-19 groups (mild, severe) did not follow clear-cut clinical symptoms. Moreover, can the authors exclude that all individuals assigned to the no Covid-19 group were really healthy controls? It is well-known that a high percentage of infected individuals do not develop the classical Covid-19 symptoms.

Author Response

Reviewer 4

n this study the authors aimed at exploring the impact of COVID-19 infection on the lipid profiles of the follicular fluid to identify biomarkers with prognostic potential. They found significant differences in the follicular fluid concentrations of several lipids when non-Covid patients were compared with Covid patients that developed moderate and severe symptoms. To the best of my knowledge the reported results are novel, interesting and worth publishing. However, the ms in its present form suffers from four major limitations that should be addressed by the authors in case the handling editor invites resubmission of the suitably revised version of the ms.

  1. Lack of mechanistic focus: The paper is very descriptive and lacks mechanistic focus. The authors quantified a number of randomly selected readout parameters and performed statistical correlation analyses. They found statistically significant differences between the different experimental groups but the mechanistic basis for these differences has not been explored. The authors suggest severe Covid-19 infections induce endothelial dysfunction and hypercoagulability (this has been reported many times before by other researchers) and that these changes might impact the lipid composition of the follicular fluid. However, the authors do not explore at the molecular level how exactly systemic endothelial dysfunction and hypercoagulability (pro-thrombotic status) induces the observed alterations in the lipid composition of the follicular fluid.

Answer:

The structure of the work at the first stage included the determination of the lipid profile of the follicular fluid in patients included in the IVF program, and divided into two groups - after suffering Covid-19 and without a history of Covid-19 (the fact of suffering Covid-19 was confirmed or excluded by determining the level of IgG to SARS-CoV-2 in the blood serum is higher than the positivity index (PI)). At the second stage, we carried out a statistical analysis of the data, which allowed us to assume differences in a number of pathogenetic mechanisms with different severity of the course of the Covid-19 disease (mild or severe), echoing the data of a number of authors around the world (links in the article include 42 groups of authors) and the general knowledge of normal and pathological physiology accumulated to date. The data obtained and their interpretation proposed by us are descriptive. They are not clinical guidelines and have not been introduced into the routine practice of doctors, but allow the scientific community to use them to further study the problem of Covid-19 (and viral infections in general) in reproductive medicine and obstetrics. This is a standard way for non-targeted metabolomic approach.

  1. The ms is very difficult to read and I experienced problems when I attempted to extract and to evaluate the major results of this study. The Abstract is far to long (362 words) and does not present the major experimental findings in a concise way. Instead, it involves a number of general statements and speculations, which are not really supported by the experimental data. This is also the case for the Results and Discussion section. Except for the Mat+Meth section all other parts of the ms should be restructured and the authors should focus the attention of the readers on their experimental findings.

Answer:

Work was carried out to change the structure of the manuscript including Abstract and Summary. The sections "Results and Discussion" were also redesigned in order to increase the specification and add more in-depth analysis in the description of the results. However, taking into account the wishes of the other reviewers (we had 4 of them) about the need to expand these sections of the manuscript, we could not shorten it.

  1. Inadequate data presentation and biological relevance of the observed differences: When I look at the Fig. 1a+b I do not see major differences between the experimental groups for any of the determined readout parameter. The differences might be statistically significant but the extent of these differences is really subtle. Thus, for me it is questionable whether these differences are biologically relevant and whether or not these measures can be considered predictive readout parameters. The same is true for Fig. 2. Here again, the extent of the difference between the childbirth and the miscarriage group is very low for most of the readout parameters. Only for LPC-16:0 a difference is easily appreciated but even for this readout parameter the difference between the means is not even a factor of 2. This is also the case for the results presented in Figure 3-7. In general, statistic significance does not automatically mean that these differences are of biological relevance.

Answer:

We carried out a statistical analysis of the data, which allowed us to suggest differences in a number of pathogenetic mechanisms with different severity of the course of the Covid-19 disease (mild or severe), resonating with the data of a number of authors around the world (links in the article include 42 groups of authors) and general knowledge of normal and pathological physiology accumulated to date. This gave us the opportunity to put forward assumptions about the biological significance of the results obtained.

To date, two main types of studies are carried out by the MS method: non-targeted and targeted. The first, non-targeted, allows the simultaneous evaluation of thousands of metabolites in a sample. In off-target analysis, it is possible to determine not the absolute amounts of compounds, but the relative change between groups. They are then usually applied to hypotheses in an attempt to understand biological processes. Subsequently, they should be tested in large-scale studies. This is exactly an our case.

On the other hand, targeted analysis is hypothesis-based, i.e. it is devoted to measuring pre-specified biomarkers with acceptable accuracy (sensitivity, specificity, and area under the operating curve) to differentiate health status. In this case, a diagnostic or prognostic model can be developed using the required number of metabolites. However, it is important to note that biomarkers developed for a specific population are only suitable for that population. Thus, well-defined studies can fill relevant gaps in clinical practice in the future.

  1. Retrospective study: I am not an epidemiologist but I am a little worried about the value of such retrospective studies. Reading the Met+Meth section I learned that analysis of the lipid composition of the follicular fluid was not blinded. Moreover, assignment of the patients to the two different Covid-19 groups (mild, severe) did not follow clear-cut clinical symptoms. Moreover, can the authors exclude that all individuals assigned to the no Covid-19 group were really healthy controls? It is well-known that a high percentage of infected individuals do not develop the classical Covid-19 symptoms.

Answer:

The fact of having Covid-19 was confirmed or excluded by determining the level of IgG to SARS-CoV-2 in the blood serum above the positivity index (PI). To detect antibodies to SARS-CoV-2 in blood serum, the "Kit of reagents for the detection of class G antibodies to the SARS-CoV-2 spike protein by enzyme immunoassay" was used for the qualitative detection of antibodies in human serum (plasma) by enzyme immunoassay (ELISA). The result of the analysis was evaluated by the value of the positivity index (PI), calculated by the formula: PI = OD of the sample/Cut-off, where the OD of the sample is the optical density of the sample. The result was considered positive if the value of PI>1.2, negative if the value of PI <0.8, doubtful (uncertain) if the value of PI is in the range from 0.8 to 1.2 (Section of the article "Materials and Methods").

Thus, we are sure that the control group did not get Covid-19.

The criterion for a mild form of COVID-19 was subfebrile temperature (<38ºС) in the absence of clinical manifestations of a moderate course of infection. The criteria for the moderate form of COVID-19 were taken into account the presence of temperature above 38ºС, shortness of breath during physical exertion, signs of pneumonia with minimal or moderate lung damage (CT 1-2), the absence of clinical manifestations of a severe course of infection (Section of the article "Materials and Methods"). These diagnostic criteria are applied throughout the world and are objective. “COVID-19 Severity Classification” Clinical Management of COVID-19 OPTIONAL RECOMMENDATIONS JUNE 23, 2022 WHO/2019-nCoV/Clinical/2022.1.

The special value of the study lies in the large sample size. The work involved samples of follicular fluid collected from 237 women.

Blind analysis of the lipid composition of the follicular fluid was not performed in our study due to a number of limitations: the peculiarity of the MS method, the need to test the hypothesis, and the approved design of the study.

Round 2

Reviewer 3 Report

The manuscript has been hugely improved and can be considered for publication

Author Response

Dear reviewer, thank you very much for your decision.

Reviewer 4 Report

My comments are provided in the enclose .docx file.

Author Response

Dear Reviewer,
We have revised our article. We also moved the paragraphs you mentioned to the discussion section. We really hope that now the article can be published. We also carried out a English language correction, and slightly changed the list of references.
